



# Controls on autotrophic and heterotrophic respiration in an ombrotrophic bog

Tracy E. Rankin[1], Nigel T. Roulet[1], Tim R. Moore[1]

[1]Department of Geography, McGill University, Montreal, H3A 0B9, Canada

*Correspondence to*: Tracy Rankin (tracy.rankin@mail.mcgill.ca)

**Abstract.** Northern peatlands are globally significant carbon stores, but the sink strength may vary from year-to-year due to variations in environmental and biogeochemical conditions. This variation is mainly brought about by changes in primary production and ecosystem respiration. The processes that relate to variations in autotrophic respiration (AR; respiration by plant parts) are understood quite well, but heterotrophic respiration (HR; respiration by microbial bacteria in the soil, fungi,

etc.) is crudely measured and modelled. This will lead to biased estimates if a change favours one form of respiration over another and alters allocations of carbon to labile pools with different turnover rates. HR has only recently been shown to be more intimately linked to vegetation dynamics than once thought, particularly in wetter, oligotrophic, sedge-dominated ecosystems. The objective of this study is to determine the factors that relate to the spatial and temporal variability in respiration and its autotrophic and heterotrophic components in an ombrotrophic bog (Mer Bleue) where woody shrubs are dominant, and

to see if the more dynamic nature of HR in sedges also exists in this bog. Plot level measurements using manual chambers were used to partition respiration from both the dominant shrubs and the sparse sedges at the site, and the controls on respiration were explored by measuring a variety of environmental variables, such as air and soil temperatures (T) and water table (WT) depth. Results show that AR and HR correlate primarily with air and soil T, with WT depth playing an important role in some cases, and that a higher variability in respiration exists for the shrub plots than the sedge plots, especially when WT levels are

more variable. Our findings also show that a plant's response to changes in climate or land-use is related to different mechanisms of obtaining water resources and utilizing symbiotic relationships with other plants around them. These results will improve our understanding of peatland carbon cycling, as well as improve the conceptualization of HR.

## 1. Introduction

Northern peatlands play a significant role in the global carbon (C) cycle, covering 12% of Canada's terrestrial surface

(Tarnocai et al., 2011), and contain ~ 50% of the organic C stored in Canadian soils (Tarnocai, 2006). Slow decomposition of plant material in undisturbed peatlands leads to the accumulation of peat, making natural peatlands long term sinks of C. Following the last glaciation, peatlands have accumulated C at an average rate of 23-26 g m$^{-2}$ yr$^{-1}$ (Charman et al., 2013; Loisel et al., 2014). However, on shorter time scales, a natural peatland may be a source or a sink of C depending on the weather and environmental conditions of a given year (Dorrepaal et al., 2009; Roulet et al., 2007). Although most of the variability in $CO_2$

exchange comes from changes in gross primary production (GPP) and ecosystem respiration (ER) (Blodau, 2002; Heimann





and Reichstein, 2008), the dynamics of heterotrophic respiration (HR, part of ER) is not straight forward. HR was often considered a variable that is correlated with environmental and substrate variables (e.g. Minkkinen et al., 2007; St-Hilaire et al., 2008), but has recently been shown to be more complicated, made up of various components that are likely to change differently as environmental conditions are altered. For example, Fan et al. (2013) suggest that long-term exposure to warmer

conditions may lead to an increase in HR outpacing an increase in C input and C sequestration will weaken, which they attribute to root-soil interactions and a change in the transport of labile C. Similarly, Basiliko et al. (2012) highlight the difficulties in separating root respiration from HR. Belowground processes are more connected to aboveground production than just the slow decomposition of dead organic matter, especially when root dynamics are considered (Ryan and Law, 2005; Van Hees et al., 2005). This has been seen in sedge dominated or forested peatlands (Järveoja et al., 2018; Kurbatova et al., 2013; Wang et al,.

2014) and in permafrost ecosystems (Crow and Wieder, 2005; Hicks Pries et al., 2015). However, it is unknown whether this same level of vegetation influence on HR exists in shrub dominated peatlands as well. This paper addresses the influence of vegetation on HR in a mid-continental, raised bog.

Ecosystem Respiration dynamics have been explored in peatlands, mainly through eddy covariance techniques (e.g. Cai et al., 2010; Humphreys et al., 2014; Peichl et al., 2014) and using darkened chambers (e.g. Järveoja et al., 2018; Lai,

2012) which explore C exchange at a scale that towers cannot address. Models have been developed that attempt to predict how the components of the C balance (e.g. ER) will vary with a changing climate (e.g. Abdalla et al., 2014; Frolking et al., 2002). A peatland's response in respiration to climate or land use change has been attributed to the plant's carbon use efficiency (Lin et al., 2014), and how the carbon accumulation will be altered (Bunsen and Loisel, 2020). However, different outcomes in a peatland's C cycle following a change in climate or land use may also occur, depending on which respiration source

dominates the response. For example, a positive feedback in climate change may occur if HR dominates the increase, because the system would lose C to the atmosphere that had been stored for hundreds to thousands of years. In contrast, if AR dominates the increase, the system will either turn over newly-photosynthesizing C faster, causing a positive feedback to climate change, or may fix more C, causing a negative feedback to climate change (Hicks Pries et al., 2013). Consequently, the contributions of AR and HR to total respiration may be ecosystem specific (Griffis et al., 2000; Ojanen et al., 2012). Phillips et al. (2017)

argue that creating a large database with more robust, improved soil respiration data will benefit further developments of models that aim to incorporate terrestrial C cycling.

Additionally, it is likely that a plant's response to a change in weather conditions or following a disturbance can also be explained by the various mechanisms in which the plants obtain water resources. Malhotra et al. (2020) suggest that environmental changes, such as warming and a lowering of the water table, can alter fine root production, affecting water and

nutrient uptake and hence ER and C storage. Oke and Hager (2020) suggest that plants, in bogs especially, are influenced by the hummock-hollow topography and that the plant's distribution depends on physiological tolerances and ecological strategies. Some plants may even take advantage of the symbiotic relationships they have with other plants around them and may act as "plant-mediated HR": plants fix the respired $CO_2$ from the surrounding vegetation rather than using $CO_2$ directly from the atmosphere in the process of photosynthesis, which has been shown to be the case for *Sphagnum* mosses in some



studies (Kuiper et al., 2014; Turetsky and Wieder, 1999). This also indicates a problem in the conceptualization of ER: one cannot simply partition AR and HR contributions when there is clearly an intermediate form (plant-mediated HR) of what is traditionally thought of as HR, in that the C is assimilated from other sources, but where the rate of litter supply is related to plant production through root-soil interactions and belowground processes rather than through plant biomass. While some current models have simulated the influence of hydrological and vegetation dynamics on soil respiration (Abdalla et al., 2014;

Heinemeyer et al., 2010), most models and manual measurements only crudely partition ecosystem respiration into its autotrophic and heterotrophic components using constant ratios or fixed decomposition rates, which may lead to an overestimation of C sequestration due to unexpected allocations of C to labile pools with different turnover rates (Hungate et al., 1997). How the role of vegetation dynamics, and the more complex nature of HR, will change ecosystem structure is still not well documented.

The objectives of this study are to determine the factors that drive the spatial and temporal variability in ecosystem respiration and its autotrophic and heterotrophic components at Mer Bleue, a mid-continental, temperate, ombrotrophic raised bog. More specifically, this paper aims to 1) determine the contributions of AR and HR at Mer Bleue, 2) establish the environmental controls on AR and HR, and 3) explore the role of plant-mediated HR and the dependence of AR and HR contributions to ER on plant functional type.

**2 Methods**

**2.1 Study site**

Mer Bleue is a 28 km$^2$ ombrotrophic bog located near Ottawa, Ontario (45.41 °N, 75.52 °W). It is in a cool continental climate region, with a mean annual temperature of 6.4 °C ranging from -10.3 °C in January to 21.0 °C in July. Mean annual precipitation is 943 mm, 350 mm of which falls from May to August, with a mean annual snowfall of 223 cm (Environment

Canada; 1981–2010 climate normals). Peat depth reaches about 5 to 6 m near the centre of the bog and is shallower (<0.3 m) near the beaver pond margin. Bog development began 7100–6800 years ago, and it has a hummock-lawn microtopography (Roulet et al., 2007). The surface of the bog is covered by *Sphagnum* mosses (*Sphagnum angustifolium, Sphagnum capillifolium, Sphagnum fallax, Sphagnum magellanicum*), and the vascular plant cover is dominated by low growing ericaceous evergreen shrubs that make up about 80% of the areal coverage (mainly *Chamaedaphne calyculata*, with some

*Rhododendron groenlandicum,* and *Kalmia angustifolia*), and an occasional mix of sedges (*Eriophorum vaginatum* and *Carex Oligosperma*) (Humphreys et al., 2014; Lai et al., 2014).

The sedges have root structures that extend vertically downwards, sometimes up to 50 cm depth, and can consequently tap into the water table at deeper depths even during the drier parts of the season as well as support a greater aboveground biomass than shrubs, especially when the water table (WT) fluctuates greatly (Buttler et al., 2015; Pouliot et al. ,2012). In

contrast, the shrubs allocate more of their biomass to belowground roots, which tend to spread out laterally rather than vertically with root lengths limited to within the first 20-30 cm of the surface (Iversen et al., 2018; Murphy et al., 2009a), hence





supporting a greater belowground biomass than sedges. Shrubs also allocate energy to small, needle-like stems (small in diameter but great in height) to make use of whatever water is available to the plants in the soil, while minimizing the loss of water through transpiration (Bonan, 2008). These stems are also buried annually by the mosses, contributing significantly to

the greater belowground biomass (Murphy et al., 2009b). This seems to be true mostly for shrubs like *C. Calyculata*, the dominant shrub species at the site, while other shrubs like *R. groenlandicum* tend to have thick leaves to prevent desiccation during drought periods (Warren et al., 2021), highlighting differences in hydraulic strategies of species that can affect ecosystem function.

Although shrubs are quite adapted for relatively wet and dry conditions, with studies finding a shift to greater shrub

cover with water table draw-down (Murphy et al., 2009a), sedges are a more competitive plant functional type than shrubs, being one of the first colonizers in abandoned extracted peatlands (Lavoie et al., 2003). Although the sedges cover only 3 to 17% of the surface area of Mer Bleue (Kalacska et al., 2013), the respiration dynamics of this plant functional type is quite important. The mosses are mixed with the other vegetation, so finding plots of just mosses was almost impossible. Therefore, the plots as described below, contained either *Eriophorum* and mosses (the 'sedge section') or *Chamaedaphne* and mosses

(the 'shrub section').

## 2.2 Chamber Setup (Direct CO$_2$ fluxes)

We conducted direct CO$_2$ measurements at the plot level using manual chambers (Pelletier et al., 2007). Nine collars were in the shrub section, and nine collars were in the sedge section. All the collars were sampled weekly to bi-weekly, weather depending, from May through September in the 2018 and 2019 growing seasons.

Fluxes were obtained using a transparent static chamber (diameter of 26 cm and height of 50 cm) placed and sealed over permanent PVC collars inserted into the peat to a depth of 15 cm at each sampling location. The chamber contained a fan to allow for adequate mixing, and a cooling system was used to maintain ambient temperature conditions (Waddington et al., 2010). For each collar, a full light measurement was done using the transparent chamber, representing the net ecosystem exchange (NEE) for that plot, and a dark round was conducted using a covered chamber. This represented the ecosystem

respiration (ER) for that plot.

In the spring of both 2018 and 2019, some of the plots were manipulated to be able to tease apart the influence of vegetation (Table 1). In each area of the manual chamber set up, 3 plots were designated as reference plots, representing NEE and ER for the light and dark measurements with intact vegetation; 3 plots had all the aboveground vegetation removed under dark conditions only; and 3 plots had only the mosses removed (i.e., vascular plants remained), also under dark conditions

only. We assumed that the plots where all the aboveground vegetation was removed represented HR, with the understanding that there may have been a residual component from the decomposing roots. In the plots representing no vegetation, a root exclosure was set up and a layer of green mesh placed on top to minimize any confounding effects of temperature and moisture. We then assumed AR = ER – HR. We followed the ecosystem sign convention, where a positive NEE value represents a gain of C the ecosystem and a positive value for ER represents a loss of C from the ecosystem.



In 2018, the $CO_2$ concentrations were measured every 5 seconds over a period of 5 minutes, using an ultra-portable greenhouse gas analyser (Los Gatos Research (LGR), San Jose, California). The LGR was calibrated beforehand, and a round started when stabilized ambient concentrations of $CO_2$ were reached. In 2019, the site was too wet to safely carry in the LGR, so a smaller portable $CO_2$ gas analyser (EGM-4, PP systems, Amesbury, Massachusetts) was used instead. $CO_2$ concentrations were measured every 10 seconds for the first minute, then every 30 seconds after that, for a total of 5 minutes. The EGM-4

was zeroed before each round. In September of 2018, $CO_2$ measurements of a few collars were measured one after the other using both instruments to get a standardized set of fluxes. There was no significant difference between the fluxes measured with the two gas analysers (T = 1.59, P-value = 0.13). In both years, regression equations of concentrations over time were used to calculate a flux for $CO_2$ for each 5-minute period. Only regressions with $R^2$ values over 0.8 were kept, which resulted in less than 10% of the values being removed. There were no instances where $CO_2$ concentrations remained the same over the

measurement period, which would have indicated a very low $R^2$ value.

### 2.3 Environmental variables

        At the time of sampling, water table (WT) depth was determined manually using a permanently installed perforated PVC tube beside each set of 3 collars. Soil temperatures were obtained using a temperature probe inserted to depths of 0, 5 and 10 cm, roughly in the same location each time just outside of each collar. Daily air temperatures were obtained from the

Ottawa International Airport weather station, located about 18 km southwest of the site (Environment Canada, 2021).

        To determine if there was any hysteresis between soil water content and WT depth, continuous measurements of both variables were conducted at the meteorological station next to the eddy covariance tower about 50 m away from the manual chamber set-up. Measures of volumetric water content (VWC) at 40 cm depth were measured using time-domain reflectometry (TDR) probes (model CS615, Campbell Scientific, Alberta, Canada) inserted in the peatland hummocks, and water table levels

were determined using capacitance water level probes (Odyssey, Dataflow Systems PTY Limited, Christchurch, New Zealand). Signals from the sensors were monitored on a CR7X and a CR10X data logger every 5 seconds, averaged every 30 min (Lafleur et al., 2005) and the daily averages were used in the analysis.

        Thermocouples were installed in the peat to measure soil temperatures at 10 cm and 40 cm depths. These were measured every second, with 30-minute averages as an output. However, daily daytime averages were used in the analysis

(using excel pivot tables and filtering for values between 8AM and 6PM). Continuous 30-minute records of WT depths were also obtained in each area of the manual chamber locations, with capacitance water level probes, that were placed inside the same perforated PVC tubes previously inserted in the peat beside each set of 3 collars, as described above. Daily averages were used in the analysis.

### 2.4 Statistical analyses

Statistical analyses were performed using R statistical software ("stats" package). Simple linear and multiple regressions were conducted among the respiration fluxes (ER, HR, and AR) and the various environmental variables, followed





by multiple regression trees to confirm which environmental variables can best explain the variability in the respiration fluxes. Repeated measures ANOVA tests were conducted to determine if the fluxes from the different treatments were significantly different, and two sample t-tests were conducted to determine whether the fluxes were significantly different between the two

plant types and whether the fluxes measured with the two gas analysers were significantly different. Finally, coefficients of variation were conducted to determine the degree of variability in AR contributions to ER.

## 3 Results

### 3.1 Environmental variables

The growing season of 2018 was characterised by variable (more sporadic) weather conditions based on the manual

measurements of WT depth and soil temperatures (Soil T) at 10 cm depth, and the mean daily air temperatures (Air T) taken from the weather station nearby (Figure 1a; Environment Canada, 2021). Air temperatures ranged from 21 °C to 35 °C, soil temperatures (at 10 cm depth) ranged between 12 °C and 27 °C, and WT depth ranged between 23 cm and 47 cm depth (June – August mean WT = 34 cm depth). It was also a hot year compared to the normal averages, where the mean annual temperature for July, for example, is 21.0 °C (Environment Canada, 1981–2010 climate normals), and a drier start to the growing season

than normal for June, and July, but generally a wetter August and September than normal (Teklemariam et al., 2010), with a significant rise in WT depth following a series of large rain events.

The growing season of 2019 had less variable weather conditions than 2018, despite a greater range in WT depth; it was wetter in May and June compared to the normal averages, then consistently became warmer and drier as the growing season progressed (Figure 1b), with WT depth similar to normal averages in July and August (Teklemariam et al., 2010). Mean

daily air temperatures (23 °C to 31 °C) and soil temperatures at 10cm depth (10 °C and 18 °C) had a much smaller range than in 2018, and WT depth ranged between 20 cm and 55 cm depth (May – August mean WT = 36.5 cm depth).

A hysteresis existed between volumetric water content (VWC) and WT depth in 2018 (Figure 2a), the growing season that showed an abrupt rise in water table positions (Figure A1a). The hysteresis was not as pronounced in 2019 (Figure 2b), which had less data available for VWC and WT depth measurements, which may have led to the hysteresis being less obvious.

Nonetheless, 2019 is where water table positions more consistently decreased over the growing season and only slightly rose in September with the start of the fall rains (Figure A1), which likely also played a role in the hysteresis loop being less obvious in 2019 than 2018.

### 3.2 CO$_2$ fluxes and AR contributions

In 2018, shrub plot NEE averaged $461 \pm 103$ mg CO$_2$ m$^{-2}$ hr$^{-1}$ ($\pm$ standard deviation), averaged $195 \pm 81$ mg CO$_2$ m$^{-2}$ hr$^{-1}$ for HR, $414 \pm 154$ mg CO$_2$ m$^{-2}$ hr$^{-1}$ for ER, and $250 \pm 69$ mg CO$_2$ m$^{-2}$ hr$^{-1}$ in the "shrub only" plots (Figure 3a). Sedge plot NEE averaged $827 \pm 139$ mg CO$_2$ m$^{-2}$ hr$^{-1}$, $240 \pm 25$ mg CO$_2$ m$^{-2}$ hr$^{-1}$ for HR, $625 \pm 131$ mg CO$_2$ m$^{-2}$ hr$^{-1}$ for ER, and $356 \pm 42$ mg CO$_2$ m$^{-2}$ hr$^{-1}$ in the "sedge only" plots (Figure 3b). AR (derived from the difference between ER and HR





measurements) in the shrubs averaged $187 \pm 134$ mg $CO_2$ m$^{-2}$ hr$^{-1}$, and $385 \pm 127$ mg $CO_2$ m$^{-2}$ hr$^{-1}$ in the sedges (Figure A2 a, b), while AR contributions to ER averaged $47 \pm 24$ % for the shrubs and $61 \pm 10$ % for the sedges in 2018 (Figure 5a).

In 2019, the shrub plot NEE averaged $323 \pm 120$ mg $CO_2$ m$^{-2}$ hr$^{-1}$, $309 \pm 123$ mg $CO_2$ m$^{-2}$ hr$^{-1}$ for HR, $611 \pm 194$ mg $CO_2$ m$^{-2}$ hr$^{-1}$ for ER, and $403 \pm 135$ mg $CO_2$ m$^{-2}$ hr$^{-1}$ in the "shrub only" plots (Figure 4a). Sedge plot NEE averaged $799 \pm 176$ mg $CO_2$ m$^{-2}$ hr$^{-1}$, $426 \pm 178$ mg $CO_2$ m$^{-2}$ hr$^{-1}$ for HR, $729 \pm 218$ mg $CO_2$ m$^{-2}$ hr$^{-1}$ for ER, and $323 \pm 107$ mg $CO_2$ m$^{-2}$ hr$^{-1}$ in the "sedge only" plots (Figure 4b). AR fluxes in the shrubs averaged $378 \pm 164$ mg $CO_2$ m$^{-2}$ hr$^{-1}$, and $343 \pm 142$ mg $CO_2$ m$^{-2}$ hr$^{-1}$ in the sedges (Figure A2 c, d), while AR contributions to ER averaged $62 \pm 16$ % for the shrubs and $55 \pm 14$ % for the sedges
(Figure 5b).

**3.3 Statistical analyses**

Repeated measures ANOVA show that the fluxes from the different manipulation treatments were significantly different for both the sedges ($F = 24.4$, $P = 0.00039$) and the shrubs ($F = 6.045$, $P = 0.0077$) in 2018 as well as the sedges ($F = 4.9$, $P = 0.018$) and the shrubs ($F = 4.57$, $P = 0.021$) in 2019. Between the vegetation types, ER, and respiration from the plots
with the mosses removed had a significant difference between the sedges and the shrubs (p-values $< 0.1$), but only for 2018, and not for 2019, whereas NEE was only significantly different between the sedges and the shrubs in 2019 and not 2018 (p-value $< 0.05$). Subsequently, between the two years, NEE, ER and respiration from the shrub only plots were significantly different (p-values $< 0.05$), but not for the sedge only plots.

Both the variance in fluxes of ER and HR were correlated with air and soil temperatures when the environmental
controls on the $CO_2$ fluxes were considered individually. The variance in AR fluxes was a bit more complex. The growing season of 2018 showed no relationships with AR fluxes for any of the environmental variables, whereas for 2019, WT depth explained much of the variance in the shrubs, and air and soil temperature much of the variance in the sedges (Table 2). Correlation analyses (not shown) revealed a positive relationship between temperature and respiration, where warmer temperature increased ER and HR for example, whereas a negative relationship was revealed between WT depth and
respiration, where a lower WT increased ER and HR.

Multiple regression analyses though, showed the interactive effect of both temperature *and* mater table position explained much of the variance in $CO_2$ fluxes for ER and HR. This was true for both plant types and in both growing seasons. However, there were only strong relationships found in the sedge plots between AR and a combination of WT depth and air temperature in 2018, and between AR and a combination of WT depth and soil temperature in 2019 (Table 3). The fluxes from
the shrub plots showed no relationship between AR and the environmental variables with multiple regressions in either growing season.

Nonetheless, the sporadic rain events in the growing season of 2018 resulted in quite variable AR contributions to ER from the shrubs, with a coefficient of variation of 54%, whereas the AR from the sedges only had a coefficient of variation of 19%. In contrast, the variation in AR contributions to ER in 2019 was much less variable for the shrubs, with a coefficient of
variation of ~30% for both plants (all coefficients of variation were significant with $R^2$ values ~ 0.90 and p-values $< 0.001$).



Although, if one were to remove the one very low AR contribution value from the shrub time series, which occurred at the hottest and driest part of the season, the average AR contribution for the shrubs in 2019 would be much greater than the sedges (~70%) and even less variable.

## 4 Discussion

### 4.1 CO$_2$ fluxes and environmental controls on AR and HR

The objectives of this study were to partition the respiration fluxes at Mer Bleue into its autotrophic and heterotrophic components and explore the dynamic component of HR. We will start in the next section by discussing the environmental controls on the respiration fluxes at Mer Bleue and delving into the contributions of AR and HR. Then we will end with a discussion of the dynamic nature of HR by exploring the role of plant-mediated HR and the dependence on plant functional type to AR and HR contributions.

ER and NEE were similar to those found in other studies (Bubier et al., 2007; Flanagan and Syed, 2011; Humphreys et al., 2014; Sulman et al., 2010), where the sedge plots showed greater respiration and NEE fluxes than the shrubs plots (Helbig et al., 2019; Lai, 2012). We found that average AR contributions to ER at Mer Bleue, calculated from direct plot measurements, were also consistent with findings in the literature (Maier and Kress, 2000; Schuur and Trumbore, 2006). Moore et al. (2002) for example, estimated that HR contributed about 46% to total ER at Mer Bleue by using a peatland decomposition model, but the study used values reported in the literature for base metabolic rates and temperature effects, not direct observations. In contrast, Stewart (2006) found HR to be about 63% of ER, using plastic leaves to try to maintain microclimate where the shrubs were removed, but was unable to establish strong links to any controls on the respiration components. Although, Hardie et al. (2009) also reported AR contributions from a blanket bog in the UK uplands to range between 41% and 54% of the total ecosystem CO$_2$ flux, and they used direct static chamber measurements.

The respiration fluxes varied, sometimes considerably, and our results show that the variability in ER and HR was driven by changes in temperature and WT position. For example, air and soil temperatures had the greatest influence on CO$_2$ fluxes, especially for measures of ER and HR when linear regressions were conducted with individual environmental variables (Table 2). While in some studies, it may seem as though temperature is the dominant factor driving changes in ecosystem functioning and peatland C cycling (Cai et al., 2010; Charman et al., 2013), others indicate that soil moisture (or the degree of wetness) may also play an important role (e.g. Belyea and Malmer, 2004). Von Buttlar et al. (2018) suggest that together, heat and drought events lead to the strongest C sink reduction compared to any single-factor extreme. Mäkiranta et al. (2010) similarly state that a warming climate may raise respiration from peat decomposition, but only if the decrease in moisture of the surface layers is minor, thus favouring further decomposition.

Multiple regression analysis shows it was the interactive effect of both temperature and water table position that explained much of the variance. This was especially true for the sedges (Table 3). These findings are partially explained by the change in weather conditions and the functioning of the plants themselves. Where the growing season of 2018 was





characterised by a sharp rise in WT mid-way through the season and consisted of a hotter and drier June and July than normal,
the growing season of 2019 was characterised by less variable weather conditions, but more wet in May and June than normal
(Figure 1). Considering that sedges can tap into deep water sources, it is reasonable that the respiration of the sedges would be
more affected by water table depth than the shrubs; shrub roots spread out laterally and are thus more disconnected from the
water table for large parts of the growing season, and most roots do not function well if they are in very saturated conditions
(Iversen et al., 2018; Murphy and Moore, 2010). The HR fluxes seem to follow the same general trend as the ER fluxes for the
shrubs, more so than for the sedges, in both years despite the more variable weather conditions in 2018; possibly highlighting
the stronger influence of soil temperature than WT depth on respiration fluxes for the shrubs. Furthermore, the ER and dark
respiration from the plots where the mosses were removed were significantly different between the shrubs and the sedges in
2018, as well as significantly different between the two study years for the shrubs. This would further suggest that sporadic
weather conditions and fluctuating WT depth has more of an effect on the respiration from the shrubs than it does on the
sedges.

Temporal and spatial variability in respiration arise because AR and HR are affected differently with climate
variability and land-use. Wang et al. (2014) suggest that both HR and AR are affected by changes in air temperature, but that
HR does not acclimate as fast as AR, so we often see a shift towards higher HR/AR ratios in warming experiments. For
example, Grogan and Jonasson (2005) found that newly-photosynthesized C by plants was more sensitive to changes in
temperature than the C derived from older stores of soil organic matter deeper (SOM) in the peat. AR contributions to ER were
highest in cooler and wetter conditions and lowest in hotter and drier conditions and varied considerably, especially in 2018
(Figure 5). The erratic behaviour in weather conditions throughout the growing season of 2018 may explain the lack in any
detectable statistical relationship relating AR to the environmental variables in the shrubs. However, AR showed a statistically
significant relationship with WT depth in 2019 for the shrubs, highlighting the threshold of where the WT may get disconnected
from surface processes; the WT dropped even further towards the end of the growing season in 2019 than it did throughout the
growing season of 2018. It seems that the less variable weather conditions, and increased wetness towards the beginning of
the growing season, may have led to both plant types having a similar AR contribution in 2019.

### 4.2 Plant mediated HR and dependence on the plant functional type

One of the keys to understanding how the vegetation responds to the surrounding environment is to determine the
capacity of the plant functional types to adapt to hydrologic and temperature extremes, or hot and dry conditions (Porporato et
al., 2004). The sedges have much higher productivity rates than the shrubs for this reason (Frolking et al., 1998); the vegetation
not only possess roots that can survive in semi-permanent saturated conditions, but also tend to allocate a lot of their energy
to aboveground leaves to increase the loss of water to the atmosphere and balance the presence of an increased water supply.
Sedges have vertical root structures that can tap into the WT at deeper depths even during the drier parts of the season (Buttler
et al., 2015) and can consequently support a greater aboveground biomass when WT depth fluctuates, hence showing a higher
average AR contribution to ER than the shrubs in 2018 (Murphy et al., 2009a). On the other hand, shrubs, which often dominate





ecosystems like bogs that have a water table at a greater depth for longer periods of time, allocate more of their energy to belowground roots and to smaller needle like stems so they can make use of whatever water is available to the plants in the soil, while minimizing the loss of water aboveground through transpiration (Bonan, 2008; Murphy and Moore, 2010). The shrubs seem to take advantage of this, by relying on the water retained by the mosses closer to the surface (Nijp et al., 2017),

and hence show a greater variability in aboveground respiration and consequently in AR contributions to ER when the WT depths fluctuate a lot like they did in 2018 (Mccarter and Price, 2014). It also possibly explains why AR contributions to ER are greater for the shrubs than the sedges in 2019, when changes in WT depth were more consistent.

The respiration dynamics depend on the mechanisms of the different plant functional types in obtaining water resources, and the relationships of the vascular plants with the mosses seem to play a vital role in how the plants respond to a

change in climate. Indeed, Järveoja et al. (2018) found in a fen in northern Sweden, that it was plant phenology that drove respiration dynamics rather than abiotic factors. In the shrubs, the $CO_2$ fluxes were, at times, greater for HR than they were in the shrub only plots. These instances seem to coincide with periods that were hot and dry (Figures 3 and 4), and in 2018, was a phenomenon only seen in the shrubs; the sedges never showed this despite measurements taken around the same time. This suggests that the shrubs have a more intimate symbiotic relationship with the mosses around them than do the sedges, as

Chiapusio et al. (2018) found. The multiple regression trees show that in 2018, air temperature was the factor that best predicted the $CO_2$ fluxes for the sedges (importance of ~ 70%) followed by WT depth (importance of ~ 30%), whereas soil temperature best predicted the $CO_2$ fluxes for the shrubs (importance of ~ 50%) followed by air temperature (importance of ~ 40%), for both ER and AR (all $R^2$ values ~ 0.70). Air and soil temperature seemed to be the best factors to predict HR for both plant types, with a combined importance greater than 80% ($R^2$ ~ 0.80). This, along with a more pronounced hysteresis loop in 2018,

supports our argument that the shrubs are more disconnected with the WT dynamics than the sedges. A change in soil temperature, which affects mainly the surface would influence the shrub's response more so than WT position or soil moisture, whereas the sedges would be more affected by changes in WT depth for most of the growing season. In 2019, on the other hand, DOY 191 – 217 was one of the hotter parts of the growing season, where the water tables during this hot period were lower than they were for the dry period in 2018, and consisted of less sporadic rain events, indicated by the less obvious

hysteresis loop. Results from multiple regression trees show that WT depth was a much more important factor in predicting the resulting $CO_2$ fluxes in 2019, with an importance of ~ 40% in most cases (all $R^2$ values ~ 0.60). Air temperature was still the more prominent factor though, with an importance of ~ 60% in most cases ($R^2$ ~0.70). This may explain why the $CO_2$ fluxes were, at times, greater for HR than they were in the shrub only and sedge only plots in 2019. These findings could indicate that both vascular plants have some sort of relationship with the mosses, as Crow and Wieder (2005) found in their

study, or it could be explained by the ability of the mosses, with their "phenotypic plasticity," to cope with rising temperatures and repeated droughts (Jassey and Signarbieux, 2019).

There were not enough datapoints in 2018 to test the statistical significance of this, but Lai et al. (2014) found the relationship with temperature changed with varying moisture conditions. In 2019, Multiple regression analyses in the shrubs showed stronger relationships between ER and HR and air temperature and WT depths when the water table was above 35 cm



($R^2$ increased to above 0.90 in both cases, with p-values $< 0.05$), whereas the relationships broke down when the water tables were below 35 cm. The opposite trend was shown in the sedges, where the relationship between HR and air temperature and WT depths was greater when the water table was below 35 cm ($R^2$ increased to 0.99 with a p-value $< 0.05$). We assume the same was true for the relationship with ER, although there were not enough datapoints with higher WT depth to confirm this. This is one limitation of our study, where more continuous measurements of the controls on respiration and its components

would be beneficial.

    Similar manipulations have been applied to chamber set ups to determine contributions of AR and HR by removing all of the roots belowground as well, a process known as girdling (Hahn et al., 2006; Hardie et al., 2009). However, these were done mainly in forested systems where roots are more easily removed without disturbing surrounding vegetation like the mosses surrounding the vascular plants in a bog. In peatlands, this is too invasive an approach, and we opted to remove only

the aboveground vegetation, while keeping in mind that residuals of the roots left behind may contribute to the fluxes we measure. There was, at times, a difference in respiration between the light and dark rounds measured from the clipped plots, especially in 2018 (data not shown). This may be explained by the slow decomposition of the roots, especially in the sedges, where constant re-clipping throughout the growing season was necessary. Stewart (2006) suggests, for example, that the soil organic matter decomposition is 1.6 to 1.9 times greater in the hollows (where the sedges mostly reside) than in the hummocks.

Marinier et al. (2004) found that re-clipping was necessary in their study, but that a root exclosure helps in minimizing the ingrowth of new roots; thus, we also included a root exclosure around our plots. This re-clipping requirement may also explain why the repeated measures ANOVA analyses between the treatments was not as significantly different in 2019 (p-value $<$ 0.05) than in 2018 (p-value $< 0.01$ for the shrubs and p-value $< 0.001$ for the sedges). However, we also did not find any statistically significant difference between the HR fluxes between the plants, which one would expect if a difference in root

residuals were to play a major role. This finding was promising; respiration from all the plots without vegetation were showing similar values throughout the growing season. There was also no difference between the two years as well though, which was more surprising as the WT depths seemed to have some influence on the HR fluxes, especially when considered alongside the dominant effect of temperature.

    While the remnants of roots in the clipped plots may partially explain why the respiration values were sometimes

higher in these plots than in the shrub only or sedge only plots, we cannot ignore that this phenomenon occurred mostly when it was hotter and drier. Zeh et al. (2020) for example, found a higher degree of decomposition of peat under sedges than under shrubs, particularly when temperatures were higher. It may also be possible that the vascular plants in these conditions are inhibiting the respiration of the microbes below (Robroek et al., 2016), with the mosses providing a priming effect to respiration (what may be called plant-mediated HR). For example, Gavazov et al. (2018) found enhanced heterotrophic decomposition of

peat carbon due to rhizosphere priming, and Basiliko et al. (2012) similarly suggest that a priming effect may occur when decomposition of soil organic matter is stimulated by rhizodeposition. In our case, the mosses may be assimilating C from the roots of the vascular plants and release that back to the atmosphere as another source of respiration in addition to that which is derived directly from photosynthesis (Turetsky and Wieder, 1999). Metcalfe et al. (2011) also suggest that the amount of C

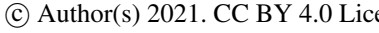


allocated belowground is governed by the total amount of C acquired by photosynthesis, which is likely to be higher for plants
that have both a greater leaf area and higher photosynthetic rates. In our study, it is clear in both growing seasons that NEE
was higher in the sedges than it was in the shrubs. So, when the mosses were removed, they were no longer able to provide
this priming effect, and when weather conditions became warmer and drier, the clipped plots, which represent HR, released
more $CO_2$ than the plots that only contained the vascular plants. A further look into the link with belowground processes may
help support this claim.

**5. Conclusions**

ER and HR seem to be primarily related to air and soil temperature for both plant types and for AR in the sedges,
however, interactive effects of environmental variables occur, with WT depth playing a significant role in some cases.
Additionally, there is some plant dependence on the dynamics of respiration, with the shrubs showing more variable respiration
values and potentially having a greater relationship with the mosses than do the sedges. This study provided a detailed analysis
of partitioning ER, especially with regards to unveiling the presence of the intermediate form of respiration we deemed plant-
mediated HR and has furthered our knowledge of C cycling in peatlands.

Given the complex nature of respiration and its components, it is clear that future studies should consider obtaining
more continuous measurements of respiration fluxes, through an automatic chamber set up for example, and that belowground
resources are seemingly quite significant to understanding respiration (e.g. root dynamics). Hungate et al. (1997) found in a
grassland that if demands for both water *and* nutrients are not met, this will lead to a higher loss of C from plants through root
turnover, respiration, and exudation. Thus, we suggest an in-depth exploration of pore water analyses, through measures of
dissolved organic carbon, and nutrients, such as phosphorus and nitrogen, will be helpful. Tools such as root exudate analyses,
and stable and radioactive isotopes have been used more frequently over the last few decades to determine the source of
respired C (Hahn et al., 2006; Hardie et al., 2009), analyses for which we suggest this project would also benefit.





**Appendix A**

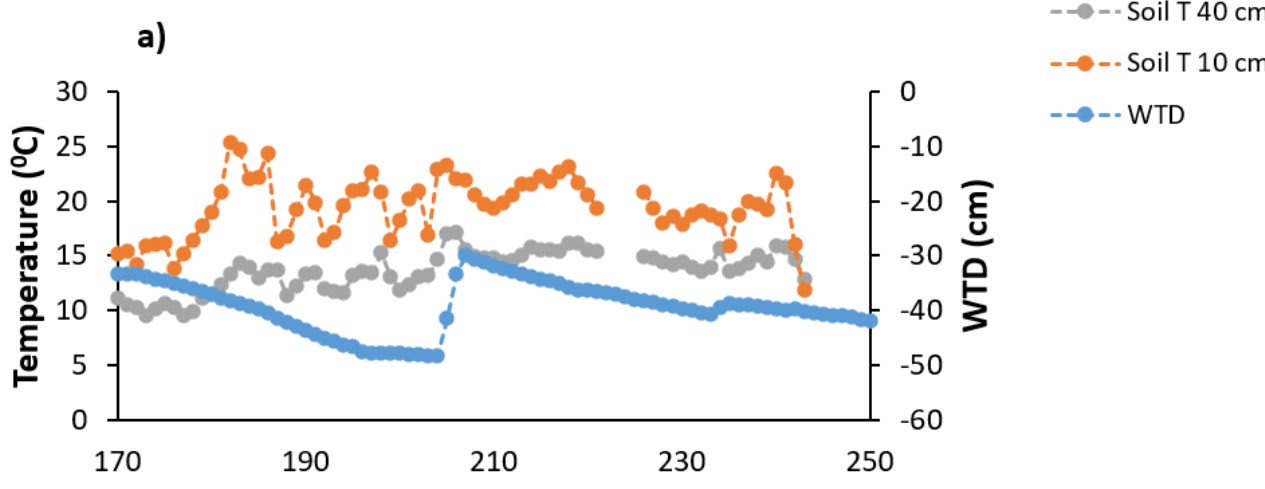

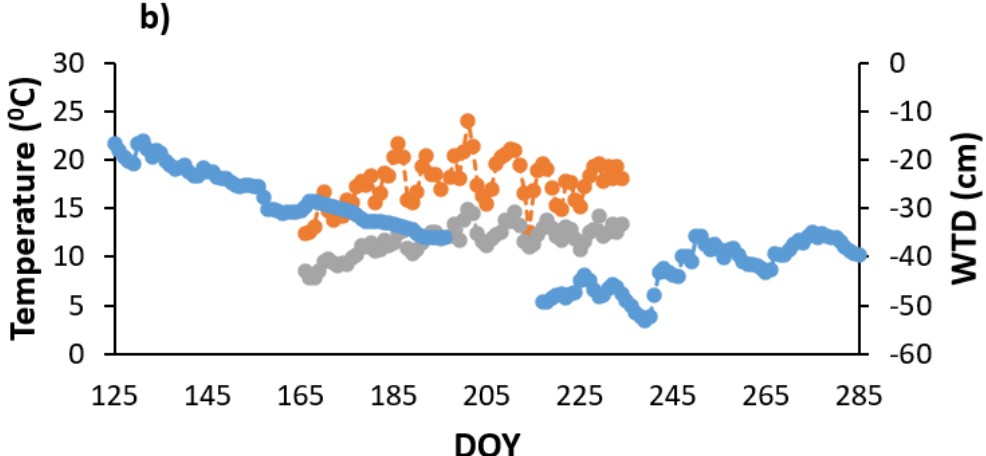

**Figure A1. Continuous measures of soil temperatures (Soil T) at 10 cm and 40 cm and water table depth (WTD) for the growing seasons of a) 2018 and b) 2019 derived from the eddy covariance tower near the manual chamber set up.**





**Figure A2. Average CO₂ fluxes in the a) shrub plots and b) sedge plots across the growing season of 2018, and CO₂ fluxes in the c) shrub plots and d) sedge plots across the growing season of 2019.**



**Author Contribution**

Tracy E. Rankin designed the experiments, with the support of Nigel T. Roulet, and carried them out. Tracy E. Rankin also
prepared the manuscript with contributions from all co-authors.

**Competing interests**

The authors declare that they have no conflict of interest.

**Acknowledgements**

This study was funded through grants from the Natural Sciences and Engineering Research Council of Canada (CRD
programs). The authors gratefully acknowledge the support from M. Dalva, M. Hassa, K. Hutchins, Z. Humeau (McGill) as
well as from Elyn R. Humphreys (Carleton University) for the environmental data from the eddy covariance tower.

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



**Table 1. Manual chamber set up with descriptions of manipulations and reported measurements.**

| Measurement | Plot Manipulation | Measurement method (Direct - DT, Derived - DV) |
|---|---|---|
| ER / reference plots | Full vegetation: shrub + mosses and sedge + mosses | DT; dark conditions, average of triplicates |
| HR / clipped plots | All aboveground vegetation removed; both shrub and sedge sections | DT; dark conditions, average of triplicates |
| NEE / reference plots | Full vegetation: shrub + mosses and sedge + mosses | DT; light conditions, average of triplicates |
| "Shrub Only" plots | All mosses removed, only shrubs remain | DT; dark conditions, average of triplicates |
| "Sedge Only" plots | All mosses removed, only sedges remain | DT; dark conditions, average of triplicates |
| AR | N/A | DV; ER - HR of averaged triplicates |






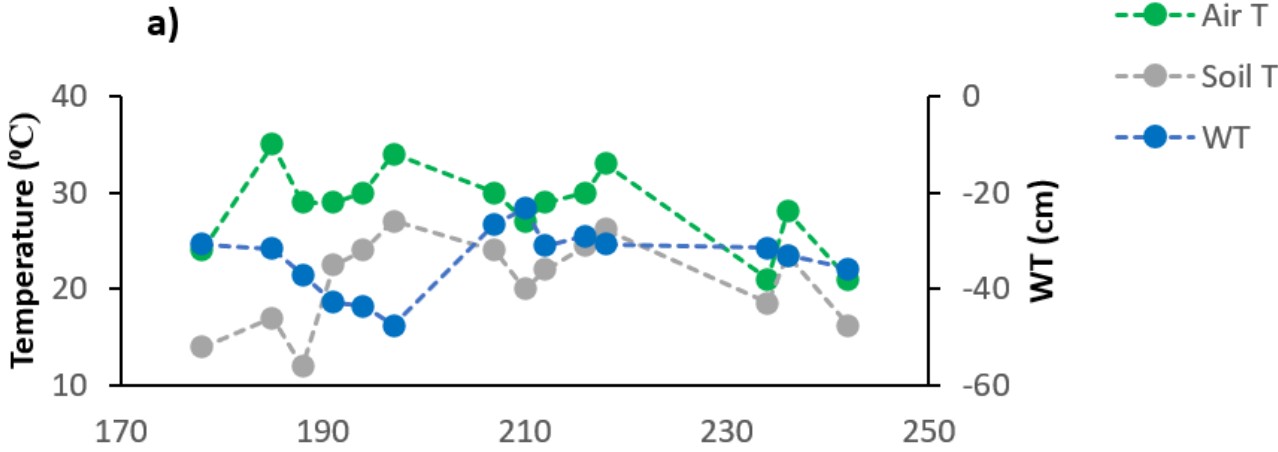

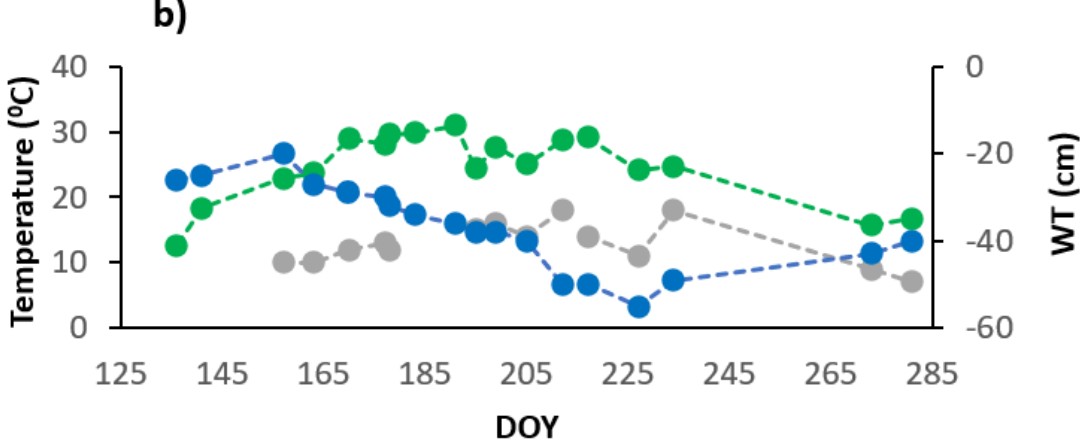

**Figure 1. Environmental variables for the growing seasons of a) 2018 and b) 2019. Soil T is soil temperature at 10 cm depth, taken manually along with WT depth, while the mean daily air temperatures (Air T) were taken from the weather station nearby.**



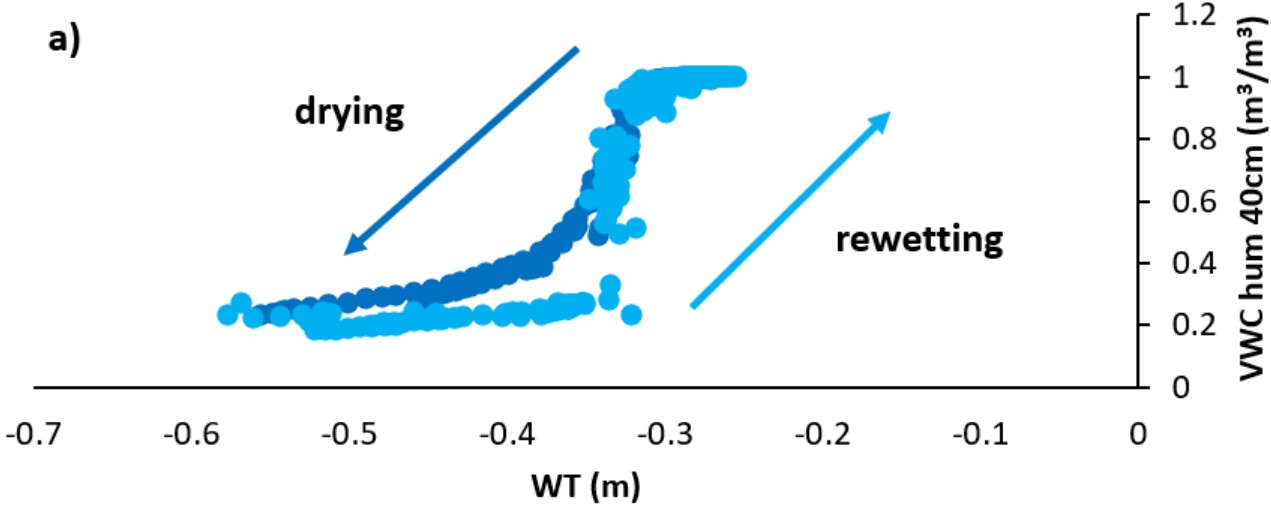

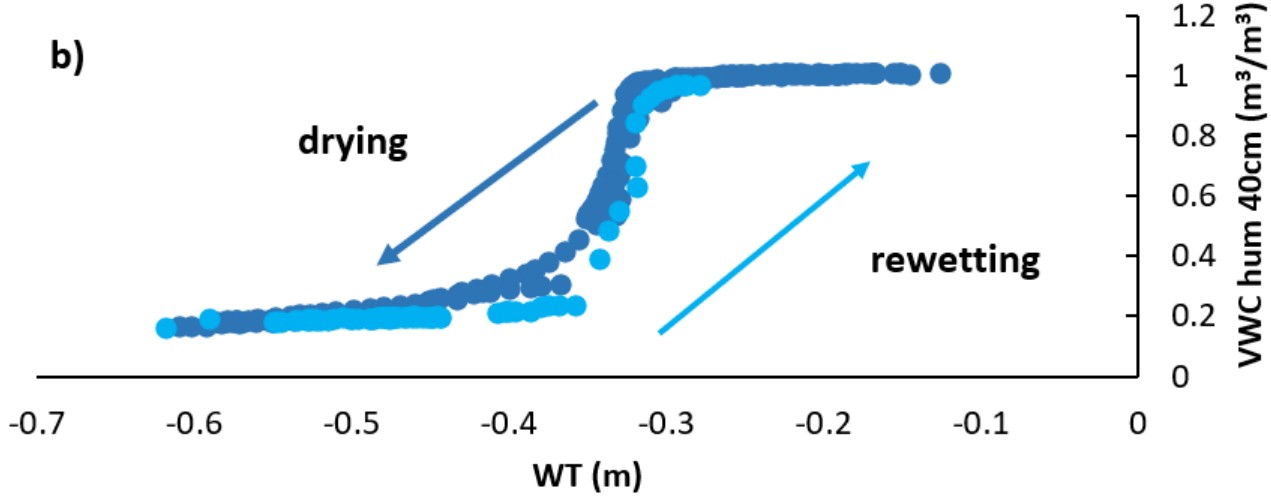

Figure 2. Hysteresis a) in 2018 and b) in 2019, between WT depth (m) and volumetric water content (VWC, $m^3/m^3$) at 40 cm depth in the hummocks.



**Figure 3. Average CO₂ fluxes in the a) shrub plots and b) sedge plots across the growing season of 2018 (± Standard Error).**





**Figure 4. Average CO₂ fluxes in the a) shrub plots and b) sedge plots across the growing season of 2019 (± Standard Error).**



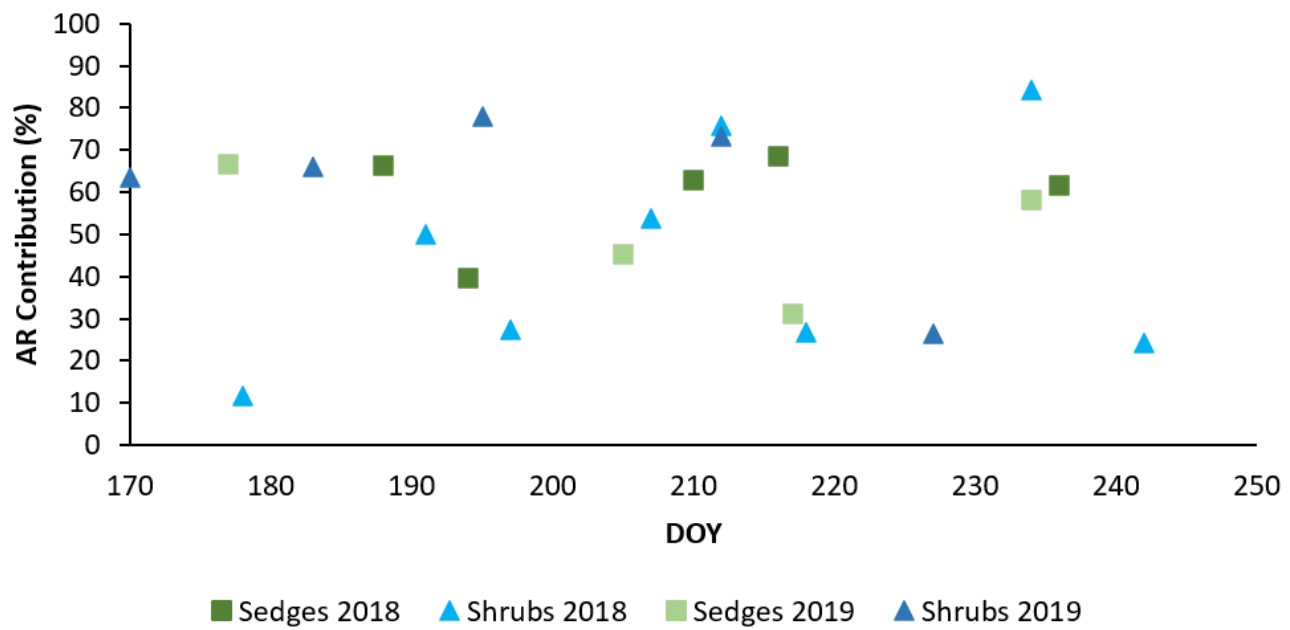

**Figure 5. AR contribution (%) to ER across the growing seasons of 2018 and 2019.**


**Table 2. Coefficient of determination ($R^2$) for linear regressions between respiration (ER, AR, and HR) and environmental variables in 2018 (bold) and 2019 (italics). WT is water table depth, Air T is air temperature measured at the flux tower, Soil T is soil temperature at 10 cm depth. Relationships are significant at * P<0.1 and ** P<0.05; All blank entries are not significant.**

| Environmental Variable | Shrubs | | | Sedges | | |
|---|---|---|---|---|---|---|
| | ER | HR | AR | ER | HR | AR |
| Air T | **0.57**\*\* | *0.74*\*\* | | *0.63*\*\* | *0.44*\* | *0.52*\*\* |
| | *0.72*\*\* | **0.74**\*\* | | **0.59**\*\* | **0.73**\*\* | |
| Soil T | **0.64**\*\* | **0.58**\*\* | | *0.55*\*\* | **0.63**\* | *0.61*\* |
| WT | | | *0.46*\*\* | | *0.62*\*\* | |






**Table 3. Coefficient of determination ($R^2$) for multiple regressions between respiration (ER and HR) and environmental variables in 2018 (bold) and 2019 (italics). WT is water table depth, Air T is air temperature measured at the flux tower, Soil T is soil temperature at 10 cm depth. Relationships are significant at * P<0.1 and ** P<0.05; all blank entries are not significant.**

| Environmental Variable | Shrubs | | Sedges | | |
|---|---|---|---|---|---|
| | **ER** | **HR** | **ER** | **HR** | **AR** |
| **Air T + Soil T** | **0.51\*\*** | **0.71\*\*** | | **0.71\*** | |
| **Soil T +WT** | **0.55\*** | **0.49\*\*** | | | *0.97\*\** |
| | | *0.64\** | | | |
| **Air T + WT** | **0.46\*** | **0.68\*\*** | **0.77\*\*** | **0.63\*** | **0.82\*** |
| | *0.75\*\** | *0.9\*\** | *0.58\*\** | *0.67\*\** | |
| **Air T + Soil T + WT** | | **0.69\*\*** | **0.7\*** | | *0.97\*\** |
| | | *0.85\** | | | |