# Peer review of "Controls on autotrophic and heterotrophic respiration in an ombrotrophic bog"

_Biogeosciences, 2021_

## Author Response (AR1)

**Replies to Reviewer Comments**

Reply to Editor

Both reviews made a range of positive suggestions and I think you have proposed a range of important revisions. I am recommending major revisions in light of their comments. In particular, I would like you to further consider the point R1 makes about treating the years as a single dataset. If you consider them to be entirely different, I think you should demonstrate this via some statistical test and/or motivate the text better so the reader is clearer on this analysis. For me, while I have no issue in itself with analysing the years separately, it does raise questions about how general our interpretations can be as the entire dataset is only two years long and variability is marked. This needs to be carefully addressed in revision.

*We thank the editor for their suggestions. With regards to treating the dataset as two separate years: 2018 was an anomalously hot year, and the resulting respiration fluxes should be considered as two separate populations. Combining the two years of data would have led to spurious relationships with the environmental variables. One can see this in the supplemental figures added, where for the shrubs especially, respiration was much higher in 2019 for the same temperature even though the slopes and $R^2$ values were similar. This was true for relationships between respiration and both soil and air temperature. Although the relationships with temperature were more similar between the years for the sedges, the results would not have been comparable with the shrubs if the dataset came from a different timeframe. So, we decided to keep the analyses of the two years separate.*

*The main revisions addressed the concerns of the reviewers regarding the statistical analyses, but we address each question / comment in turn below.*

Reply to Reviewer 1

This study used field measurements of CO2 fluxes from control and vegetation removal plots to estimate ecosystem respiration, heterotrophic respiration (HR), and autotrophic respiration (AR) in an ombrotrophic bog ecosystem over two growing seasons. The study analyzed the correlations of temperature and water table with respiration fluxes for the two years. The sensitivity of different respiration fluxes to environmental factors is an important question with implications for understanding ecosystem carbon flux responses to changing climate, as is well explained in the Introduction. I thought the study was well designed and produced a valuable dataset for understanding these fluxes and their controls in bog ecosystems.

*Thank you for your comments.*

In my opinion the statistical analysis portion of the study had some weaknesses that could be addressed.

*We will address each of the comments.*

First, some of the statistical methods are not explained in enough detail in the methods section. In particular, it's not clear how the "multiple regression trees" were conducted or how this method was defined. A full explanation and/or citation for that method would be helpful.

*In the revised manuscript, we included more detailed explanations of the statistical methods used along with citations of other studies that have used these methods (page 6).*

Second, the statistical methods rely on linear regressions. Moisture interactions with respiration in particular are often nonlinear (a threshold dependence is suggested in the Discussion, for example) so I would recommend testing whether linear relationships are an appropriate model for the processes of interest and, if not, applying nonlinear methods where appropriate.

*We have discussed in the revised manuscript the linear relationships of respiration fluxes with air temperature and soil temperature and have showed this by including the scatterplots made in the correlation analyses (added as supplemental material) as well as a table that includes the correlations in the appendix. Although there does not seem to be any relationship with WT depth when all the data points are considered, linear or otherwise, there are linear relationships over a certain range of data that coincide with what others have found in the literature, and which was discussed in the original manuscript (see supplemental figures).*

Third, it's not clear why the two years were analyzed separately instead of combined as a single dataset. Since it was all the same site and treatments, it would make sense to treat the whole time series as a common dataset and potentially this would give the overall statistical analysis more power. While it is interesting to see if some relationships differed across years, I think a good default assumption would be that the site should behave similarly in different years unless there is a compelling reason to expect otherwise. I suggest conducting the statistical analysis for the whole dataset across both years and perhaps contrasting those results with analyses for individual years if there are significant differences.

*2018 was an anomalously hot year, and the resulting respiration fluxes should be considered as two separate populations. Combining the two years of data would have led to spurious relationships with the environmental variables. One can see this in the supplementary figures where for the shrubs especially, respiration was much higher in 2019 for the same temperature even though the slopes and $R^2$ values were similar. This was true for relationships between respiration and both soil and air temperature. Although the relationships with temperature were more similar between the years for the sedges, the results would not have been comparable with the shrubs if the dataset came from a different timeframe. So, we decided to keep the analyses of the two years separate (as responded to the editor).*

Finally, the results of the statistical analysis that are present are very limited. Only statistical significance metrics, coefficients of variation, and R2 values are shown. This means that the manuscript never reports the direction or slope of the linear relationships and therefore leaves out a lot of potentially useful information. Statistical significance measures on their own are much less informative if they are not matched with information on how the relationships actually

looked. I would recommend at minimum including the linear regression parameters (slope and intercept) in a table. Even more useful would be scatter plots with regression lines showing the data and fit relationships for fluxes and environmental factors (especially if some of the relationships were particularly interesting or significant). Overall, it seems like the study generated a useful dataset but did not fully analyze it.

*We added scatterplots a supplemental material to show the relationships with the environmental variables, along with a table of correlations in the appendix to show the strength of the relationships. We also added individual p-values in Tables 2 and 3 next to the $R^2$ values, including the non-significant results, as well as p-values, degrees of freedom, and F and t statistics in the text for the ANOVA and sample t-test results.*

Other comments:

Line 63-68: This explanation of "plant-mediated HR" did not make sense to me. First it is explained as plants fixing carbon that was recently respired from surrounding vegetation. This isn't HR, it's reabsorption of respired CO2. And I don't see why this is a problem for calculating ER. From the perspective of ecosystem carbon balance, it shouldn't matter if the carbon source for photosynthesis came from ecosystem respiration or from the atmosphere — aren't they all carbon molecules in the end? Does it make a difference how far they traveled? Later, plant-mediated HR is explained as having to do with root-soil interactions and litter supply, which seems like a different issue from reabsorption of respired CO2. A different process that could be called "plant-mediated HR" is supply of C to the rhizosphere that is immediately respired by heterotrophic organisms. This explanation is more consistent with the Discussion paragraph on this topic, which is mostly about rhizosphere priming effects. This does seem like an issue for partitioning AR and HR because it is plant-supplied C that would be cut off by removing plants but it is not strictly AR. But this does not fit with the explanation of "plant-mediated HR" in the Introduction text.

*We explained that there are three sources of $CO_2$ belowground for which we cannot discriminate: $CO_2$ that is supplied as a substrate by the vascular plants (priming effect), root respiration itself, and heterotrophic respiration by microbial bacteria, etc. that is not associated with the roots.*

*Instead of using the term "plant-mediated HR", we discussed respiration more as an association of $CO_2$ with the structure of the peat. For example, with regards to the mosses, we have recycled C as $CO_2$ that is fixed by the mosses to be used in photosynthesis. We revised the manuscript accordingly to clarify this.*

Line 123-125: The wording here sounds like the vegetation removal happened under dark conditions, but I think what is meant is that CO2 flux was only measured under dark conditions (not light conditions) in plots where vegetation or mosses were removed. Not that the vegetation removal itself was done in the dark.

*We changed the wording in the methods to make it clear that removal was done first then CO$_2$ fluxes were measured under dark conditions.*

Line 124: Plots with mosses removed are later referred to as "shrub-only plots." The same terminology should be used throughout the manuscript.

*We made sure to use the same terminology throughout.*

Line 209: The text says that ER and HR were correlated with air and soil temperatures, but based on Table 2 soil T was only significant in one year.

*We made sure to be clear in which year and for which plant type the significant relationships were found.*

Line 247: Were the influences positive or negative? And how strong? Only providing statistical significance measures and nothing else leaves out the most important information here

*Although it was mentioned in lines 213-215 of the original manuscript whether the influences were positive or negative, we agree that there was no mention of the strength of these relationships. As stated above, we included scatterplots as supplementary material as well as a table of correlations to the appendix to show the strength of the relationships.*

Line 225: Again, knowing that this interaction was significant is less useful than knowing what the relationship looked like.

*With regards to the coefficients of variation, we added that they are calculated by taking the standard deviation/mean, so they are not really associated with a significance level. However, Figure 5 was changed to include boxplots showing yearly averages to more clearly show the difference in the spread of AR contributions between the years and plant types.*

Line 265: The relative influences of soil T and water table on fluxes could be determined from the parameters of the multiple regressions rather than speculating about it based on qualitative looks from the figures as this sentence does.

*We referred back to the statistics when explaining the influences of the environmental variables.*

Line 274-275: The relative contributions of AR to ER under different conditions could be shown directly with a scatter plot of the relevant processes, or by referring to parameters of the linear regressions.

*As there are already figures of the time series of weather conditions and AR contributions as well as additional scatterplots of respiration fluxes as a function of the environmental variables, adding additional scatter plots will not add much to the paper. We did include evidence from the statistical analyses in the text to give more credence to the claims.*

Line 276-277: If there is a real statistical connection between AR and environmental drivers, then why would higher variability in environmental drivers cause the relationship to be weaker? Might this suggest that the apparent relationship is due to some other covariate that varies more slowly over the year? Or that respiration responds to environmental drivers at a particular time scale?

*With the limited sample size in AR fluxes, the relationship, as it would have been, may not have been captured properly. It may be that the respiration responds at a different time scale than our study period. The way to resolve this would be to use continuous measurements (e.g. automatic chambers), which we do not have for this study. We revised accordingly to make this clearer.*

Line 278-280: A threshold relationship with WT could be shown directly with a scatter plot of WT versus respiration. Also, a threshold response is inherently nonlinear which suggests that linear regression may not provide an accurate picture of the relationship.

*We added a scatterplots as supplementary material, but the relationships were linear within a certain range of our observed data that coincided with relationships found in the literature. Otherwise, there was no relationship with WT depth when all of the data points were considered, as stated above.*

Line 304: It seems speculative to talk about symbiotic relationships here. The data don't have enough detail to say whether there is a symbiotic component to the observed correlations.

*We no longer use the word "symbiosis" and explain instead that there is a possibility of the mosses and vascular plants having a mutual benefit to one another by their presence in the ecosystem. The vascular plants provide a source of $CO_2$ that may diffuse through the mosses, while the mosses provide moisture in the water they retain during extended periods of drought. We think the data support this possibility, but we revised the manuscript to be clear that this is a speculation and that we are not making a conclusive statement.*

Line 305-309: This should be in the results section

*We moved this part up to the results section and simply referred to the table in the text here.*

Line 315-317: This should be in the results section

*We moved this part up to the results section and simply referred to the table in the text here.*

Line 322: Wouldn't this be an interaction term in the multiple regression? The regression would indicate whether the interaction term was significant or not. And conducting the statistics across both years instead of separately by year could give better statistical power.

*Indeed, the multiple regressions would tell us whether the interaction term was significant or not, but since 2018 was an anomalous year in terms of temperature, we believe lumping the data from the two years would have only given a spurious relationship, as stated above.*

Figure 1: I think it would be helpful to superimpose continuous measurements of temperature and water table (as lines) along with the dots showing values when fluxes were measured. This would allow those time points to be placed in the context of the whole time series.

*The time series in Figure 1 shows values for the environmental variables taken at the same time as the flux measurements, so the continuous measurements were added as an appendix mainly to contextualize the manual measurements. They do correlate though, if we look at the values on the same day between the manual and continuous measurements. We decided not to add a graph to the Appendix though as we felt that this would not have added to the paper.*

Figures 3 and 4: I found these plots difficult to read with all the different colored dots. Connecting the dots with lines or plotting as bars rather than dots might make these figures easier to interpret.

*We played around with the size, colour, etc. of the figures in the revised manuscript. Hopefully they are now easier to interpret.*

Figure 5: This figure should have separate panels for the two years (similar to the previous figures) or show one long time series. Plotting them on top of each other makes the plot difficult to read.

*Figure 5 was changed to include boxplots showing yearly averages to more clearly show the difference in the spread of AR contributions between the years and plant types.*

Table 2 and 3: The bold and italics notation for different years is difficult to read, especially since the order of years is not consistent. Also, there's no reason not to show all the data. These tables should just have a line for each year (two lines per environmental variable) and show all the values (whether statistically significant or not). And, ideally, include statistics over both years of combined data. Also, the regression parameters (slope(s) and intercept) should be included.

*As stated above, we added the non-significant data as well as the individual p-values next to the $R^2$ values in Tables 2 and 3 and separated the two years of data. We felt that including all of the regression parameters here though would have made the tables too busy. We opted instead to add scatterplots of some of the relationships with slope and intercepts as labels along with a table of the correlations in the appendix.*

Reply to Reviewer 2

This manuscript presents data from a field experiment where $CO_2$ fluxes were measured in control, complete vegetation removal, and moss removal plots in an ombrotrophic bog in order to estimate ecosystem respiration. Further, the vegetation removal treatments were used to partition respiration into contributions from autotrophic respiration and heterotrophic respiration. Measurements were conducted across two growing seasons, and respiration measurements were coupled with measurements of environmental variables such as water table height and air and soil temperatures in order to identify drivers of respiration across the growing season and among different vegetation types.

While the objectives of this study and the rich dataset are valuable contributions to the field, I agree with many points made by Reviewer 1 in addressing the statistical weaknesses of this paper. I find six key points that warrant attention on behalf of the authors to improve the strength of this paper's analyses and conclusions.

*We thank the reviewer for their comments.*

The structure of the discussion is rather confusing. Perhaps separating the discussion section into environmental predictors of AR vs. environmental predictors of HR, temporal variability in AR and HR, and vegetation type differences in respiration would make for a more succinct discussion that directly relates to your manuscript's stated objectives.

*As AR is a residual term (difference between ER and HR), and AR is hence dependent on HR, we think that separating the environmental predictors of AR and HR into two sections was not a favourable option. We have re-worded some of the section headings though and moved up the last paragraph of section 4.1 to after the original line 254 as well as re-arranged some of the discussion paragraphs. Hopefully the revised manuscript flows better.*

As Reviewer 1 suggests, a clearer definition of the methods used as part of the "multiple regression trees" is necessary. Further, I suggest instead using model comparison and selection methods like stepwise AIC comparison of models to identify the suite of variables that best explain HR and AR in bog areas dominated by different vegetation types. This would better allow you to identify the most predictive combination of variables in this system.

*As was stated in the reply to reviewer 1 above, we provided a clearer definition of the statistical methods used, especially with regards to the regression trees. The authors thank the reviewer for the additional suggestions. We looked into conducting the stepwise AIC but since stepwise regressions are included in the creation of the regression trees, we felt that this would not add much to the manuscript.*

I disagree with the author's discussion of "plant mediated HR" in this manuscript. In the introduction, the author's define plant mediated HR as photosynthesis conducted using $CO_2$ respired by surrounding plants instead of $CO_2$ sourced from ambient pools. This variable is not measured at any point in this study and would require isotopic analyses of plant biomass, assuming that plant mediated HR results in significant fractionation of C isotopes so that photosynthate from plant mediated HR would bear a distinct isotopic signature than would

photosynthate from ambient sources. While the authors postulate many credible theories as to why the presence of mosses and the functional differences between shrubs and sedges might alter the physical and chemical properties that influence respiration, these ideas should instead be discussed in a section that is dedicated to describing differences in respiration among vegetation types, eliminating the rather confusing term "plant mediated HR".

*As stated in the reply to reviewer 1 comments, instead of using the term "plant-mediated HR", we discussed respiration more as an association of $CO_2$ with the structure of the peat. For example, regarding the mosses, we have recycled C as $CO_2$ that is fixed by the mosses to be used in photosynthesis. We revised the manuscript accordingly to clarify this. We also discussed that there are three sources of $CO_2$ belowground which we cannot discriminate: $CO_2$ that is supplied as a substrate by the vascular plants (priming effect), root respiration itself, and heterotrophic respiration by microbial bacteria, etc. that is not associated with the roots.*

As Reviewer 1 mentioned, the results that the authors report are compelling but insufficient to give readers a clear understanding of how the environmental variables measured here influence respiration. The tables and manuscript text should be amended to include correlation coefficients that report the magnitude and direction of the relationships analyzed in this manuscript, and all results should be reported regardless of whether or not the relationships are statistically significant. Insignificant results are interesting too! Other aspects of the tables are confusing as well. Instead of including 2018 and 2019 data in the same columns with different font faces to differentiate them, consider including separate columns for each year (unless you choose to analyze data from both years together, as suggested by Reviewer 1). I also don't understand what the second row of data under some environmental variable labels (i.e. row 2 of data in Table 2) refers to. Table structure must be amended in all tables in this manuscript to improve clarity.

*As stated above, we added the non-significant data as well as the individual p-values next to the $R^2$ values in Tables 2 and 3 and separated the two years of data. We added scatterplots of the relationships with slopes and intercepts as labels, along with a table of correlations in the appendix.*

The figures in this manuscript are often visually unclear or confusing. In Figure 2, the colors used to indicate drying vs. rewetting points are virtually identical and extremely difficult to differentiate. Perhaps change the size, color, and transparency of the points in this figure to allow readers to see differences near the asymptotes where many points are stacked on top of one another. In Figures 3 and 4, the colors for ER and NEE are also too similar to distinguish, especially when considering that the figures would be much smaller in the final published article. It is also difficult to distinguish between the blue colors used in Figure 5 for the shrub plots.

*We played around with the figures with regards to size and color. Hopefully they are clearer.*

In Figure 5, why not include error bars for AR contribution data points as the authors did in Figures 3 and 4? While connecting the points with lines across the growing season would help readers distinguish temporal trends in AR contributions among your treatments, I suggest averaging AR contributions in each plot across the growing season and then visualizing differences in AR contributions among growing season years and vegetation types using boxplots. These differences can then be verified using an ANOVA test.

*Figure 5 was changed to include boxplots of yearly averages to show the difference more clearly in means and spread of AR contributions between the years and plant types. We also added error bars to the AR values in the appendix (Figure A2).*

For Table 2, I would prefer to see panels of linear regressions that depict the relationships between respiration components and environmental variables. This table of statistical results can then be moved into the appendix.

*As stated above, we added the non-significant data as well as the individual p-values next to the $R^2$ values in Tables 2 and 3 and separated the two years of data. We added scatterplots of the relationships with slopes and intercepts as labels, along with a table of correlations in the appendix.*

An important spatial component of bogs that this manuscript largely ignores is the hummock/hollow variation in microtopography. I would suggest reframing the objectives of this study as analyzing temporal/vegetative variation in bog respiration dynamics to reflect your experimental design more accurately.

*We examined patterns of respiration in hummocks, which represent 70% of the bog (Lafleur et al. 2003), and incorporated mosses, shrubs, and sedges. We made sure to make this clear in the text.*

Specific line comments:

Line 121: How much time elapsed between the removal of plant biomass and the installation of root exclosures and the first $CO_2$ flux measurements? Were vegetation removal treatments reapplied throughout the two years of measurements?

*We explained this more clearly in the text.*

Line 182: Because hysteresis does exist to some degree, and the amount of hysteresis varies among years, why not use VWC measurements as your variable that represents soil moisture conditions instead of WT height?

*We do not have VWC measurements for the different treatments, only the data from the probes near the eddy covariance tower. We could show that the relationship between WTD and VWC are correlated though, and that WTD is thus a reasonable surrogate for changes in VWC, though it is different because of the hysteresis present. We added this explanation in the text of the revised manuscript.*

Lines 202-208: Be consistent when reporting p-values. I tend to see 3 decimal places for p-values reported, with exact values used instead of simply reporting significance thresholds.

*We added individual p-values in the tables next to the $R^2$ values (to 3 decimal places) as well as individual p-values in the text for the ANOVA and sample t-test results (since one of the values went out to 4 decimal places, we did this for all p-values reported in the text).*

Line 216: There's a small typo here, "mater" should probably be "water".

*We revised accordingly.*

Line 222: I do not think that you have the evidence to support your claim that variation in rain events (sporadic rain events) drives greater variation in AR among vegetation types. Furthermore, throughout this paragraph, you should report the coefficient of variation more accurately instead of rounding, as well as p-values and F-statistics stemming from an ANOVA that should be used to properly test the differences in AR contributions among vegetation types or among years. Furthermore, reporting your degrees of freedom associated with the F-statistic in these analyses would help the readers understand how many independent measurements are used in your analyses.

*We also added individual p-values, degrees of freedom, and F and t statistics in the text for the ANOVA and sample t-test results. Our comments on the impact of sporadic rain events were speculative and we made it clear that we are not claiming a cause-effect relationship.*

Lines 231-235: This paragraph is unnecessary given the use of subheadings in your discussion.

*We removed this paragraph.*

Lines 240-243: Perhaps remove reference to Moore et al. 2002 and Stewart et al. 2006 because these studies are not directly comparable to your results given differences in measurement methodology, which you note.

*We removed these citations.*

Line 305: When you say "importance of 70%", what is the statistic that you are reporting here, and from which statistical test is this number derived?

*We used the word "explanation" here instead of "importance". We also describe how the statistical test is derived.*

Lines 322-330: As Reviewer 1 stated in their comments, the relationship between the environmental variables and respiration components discussed in this paragraph likely stem from non-linear relationships between respiration and soil moisture in particular. Using statistical tests beyond linear regressions would be a more appropriate way to test this hypothesis.

*As stated in the reply to reviewer 1 comments, we discussed in the revised manuscript the linear relationships of respiration fluxes with air temperature and soil temperature and showed this by including scatterplots made in the correlation analyses (added as supplementary material). Although there does not seem to be any relationship with WT depth when all of the data points are considered, linear or otherwise, there are linear relationships over a certain range of data that coincide with what others have found in the literature, and which was discussed in the original manuscript (see supplementary material).*

Line 338: Other studies such as Rewcastle et al. 2020 (Pedosphere) use different methods of root exclosures that eliminate the possibility of $CO_2$ flux stemming from residual root decomposition, yet also find rather variable HR rates owing to water table and soil moisture differences irrespective of bog microtopography differences.

*The authors thank the reviewer for the suggested citation and we included it in the paper, although we specify that they are dealing with a forested bog rather than a shrub-dominated bog like Mer Bleue.*

Line 331-348: As in other sections of this manuscript, the results that you report must be more specific. Report exact p-values instead of significance, and report p-values even for insignificant results. Results from regressions should include correlation coefficients as well, and results from ANOVA tests should include F-statistics with degrees of freedom to communicate replication in your study.

*We revised accordingly as stated above.*

Line 354: My understanding of the literature surrounding bog decomposition suggests the opposite, that the high degree of secondary compounds in moss litter inhibits microbial activity, while vascular plant litter and root exudates often have a priming effect on microbial activity in bog ecosystems. Evapotranspiration surrounding vascular plants might also increase oxygen availability by lowering the water table in proximity to deeply-rooted plants, again stimulating microbial activity (further supporting the pattern observed by Zeh et al. 2020).

*We did not discuss evapotranspiration as a source of change in WT position as we did not measure this to confirm the possibility. However, the sentence here was simply mixed up by mistake with reference to the priming effect of C. We revised accordingly and both Robroek et al. (2016) and Zeh et al. (2020) found the same thing.*

Line 375: I would suggest referencing a study other than Hungate et al. 1997 that confirms this ecological principle in bogs rather than grasslands owing to the complex physio-chemical regulation of the carbon cycle in frequently water-saturated ecosystems like bogs.

*We added a citation for a more recent study (Fenner and Freeman 2011) that was conducted in a bog and the authors thank the reviewer for the suggestion.*

---

## Referee Report (RR1)

This manuscript provides a valuable dataset and analyses that describe carbon fluxes from a bog, portioned into the contributions to ecosystem respiration by autotrophic (AR) and heterotrophic (HR) sources. This dataset itself is unique, and the analyses that link respiration to the influence of water table depth, temperature, and plant community dynamics make valuable contributions to the field. The manuscript, namely the statistical analyses and figures, are much improved from the initial submission. However, I have remaining concerns regarding the interpretation of the results and the discussion section especially regarding the interactions between mosses and vascular plants that affect heterotrophic respiration rates (mainly the last paragraph of section 4.2, starting line 364).

After reviewing the original manuscript, both reviewers raised concerns with the authors' discussion of symbiotic relationships between mosses and vascular plants in relation to HR rates, particularly the ability of mosses to metabolize and respire labile carbon from vascular plant root exudates. First, it seems incorrect to discuss respiration from mosses in the context of HR, which by definition includes only respiration from the soil microbial community. Second, the ability of mosses to absorb, metabolize, and respire dissolved carbon from the soil solution is not widely accepted, and the authors give a single source from 1999 to support these speculations. The authors simply do not have the data to make claims regarding the source of the carbon respired by plants in the experimental plots studied here. I suggest instead drawing speculations regarding the effect of mosses on heterotrophic respiration rates from known relationships in the literature that connect the presence of mosses to water table depth/soil moisture and temperature, variables that the authors show significantly shape HR. It seems plausible that mosses could insulate peat from evaporative losses of water, perhaps even establishing cooler, anaerobic conditions beneath mosses that limit microbial activity and HR when moss is present. When moss is removed, the albedo of the bog surface changes so that the bog surface would become warmer and evaporation would occur more rapidly without the insulative presence of mosses to allow for more rapid aerobic microbial activity. These types of relationships, between mosses and microbial activity and the environmental factors that shape microbial activity, warrant further discussion in the manuscript segments concerning HR and might be more appropriate and scientifically sound than discussing moss respiration from dissolved carbon sources.

Other minor concerns remain as well, as noted below:

Line 9: "respiration microbial bacteria in soil, fungi, etc.": "respiration by the soil microbial community" would probably be a more concise and accurate description of HR

Line 11: "and alters allocations of carbon to labile pools with different turnover rates": The relationship between respiration and carbon substrate complexity was not something that was examined in this study and should be removed here and elsewhere in this manuscript to avoid confusing readers regarding the factors analyzed in this relationship. This argument distracts from the more evidence-based conclusions you make regarding the influence of abiotic factors and plant functional type on ecosystem respiration.

Line 20: Here and elsewhere in the manuscript, you discuss different plant water acquisition strategies as an important driver of respiration rates in this system, but what you actually discuss is differences among plant rooting structures; these differences may or may not be actually tied to water acquisition. I think you should refer to these differences as rooting structure differences instead of water acquisition differences because that more accurately describes the arguments you make in your discussion and the literature that you cite as you do not measure water acquisition in this study.

Line 31: "the dynamics of heterotrophic respiration... *is* not straightforward" should be "*are* not straightforward"

Line 32: "substrate variables": Can you be more specific? I think this covers a very broad spectrum of biogeochemical variables... Substrate quality or complexity? Substrate quantity?

Line 36: "...and $CO_2$ that is supplied as a substrate by vascular plants": This sentence is misleading, and the intended point here is unclear. $CO_2$ gas is not provided by plants as a substrate for microbial metabolism.

Line 44: "Ecosystem Respiration dynamics...": the R in respiration should not be capitalized.

Line 51: "For example, a positive feedback in climate change...": Several typos and/or confusing word choice make this sentence hard to follow and the intended point unclear.

Line 53: "... turn over newly-photosynthesizing C..." should probably be "newly-photosynthesized C"

Lines 65-68: "This also indicates a problem in the conceptualization of ER: one cannot...": This sentence is very unclear and perhaps should be broken into several separate points. It's hard to understand what connection the authors are trying to make between $CO_2$ released during HR and litter production as an intermediate contribution to HR... Carbon respired by the microbial community is still considered HR regardless of what the ultimate fate of those $CO_2$ molecules are.

Line 83: Mean annual precipitation is listed as 943 mm, but this sentence states that annual snowfall is 223 cm. Are the units for snowfall accurate? I realize that there is some discrepancy for measured snowfall and realized water increments, but are those differences an order of magnitude apart?

Line 96: "...hence supporting a greater belowground biomass than sedges": This conclusion doesn't necessarily follow logically from the information you presented in this paragraph and at the very least, deserves its own reference. Why would the shallower root structure of shrubs necessarily equate to greater below ground biomass than the more deeply rooted sedges?

Line 99: Live moss biomass still counts as aboveground biomass, so it seems illogical to consider stems buried by living moss to be considered belowground biomass.

Line 103: The conclusion made in the first sentence of this paragraph seems unsupported by the information provided. You state that sedges have a competitive advantage over shrubs, yet start this sentence by stating that shrubs thrive in both dry and wet conditions. These statements seem to contradict one another.

Line 110: Does the placement of the 9 collars represent 1 collar per plot? If so, how big are these plots within the shrub and sedge sections? As currently written, this seems to indicate pseudoreplication, where you measure respiration in 9 places with a single shrub and a single sedge plot.

Line 122: How big are each of the subplots with the vegetation manipulations?

Line 127: Were the HR plots trenched to kill roots from plants that grow outside of the plots but close enough to produce roots that grow belowground into the plot area?

Line 139: "regression equations": linear models?

Line 195: "Although it is important to acknowledge the hysteresis present...": I think you should describe *why* acknowledging hysteresis is important. It essentially means that WT depth is a better predictor of VWC in 2019 than in 2018.

Section 3.2: Listing these results in the text is rather difficult to trudge through and derive any real sense of comparison between respiration rates in different years, with different plant manipulations, and between different plant types. Lit the results in a table instead, and use this section to describe comparisons between respiration rates among your treatments... i.e. Shrub respiration was x% greater in 2018 etc.

Line 281: You mention "land-use" as a factor shaping respiration rates at many different points in this manuscript but it is certainly not a factor that you explore in this experiment... I would advocate for removing reference to land-use from this manuscript not because it's not important, but because you don't investigate the influence of land use in your study, but you do investigate the influence of environmental and vegetative properties which should be your sole focus.

Line 320: Have you considered the fact that water table differences between your vegetation types alone might explain the vegetative differences in respiration rates? When the water table is higher, more deeply rooted plants will respire in totally saturated peat conditions, and a large portion of that respired $CO_2$ will be dissolved in the soil solution instead of being released from the soil as gaseous $CO_2$. This would mean that when the water table is higher, AR is automatically lower because at least a portion of deep root respiration won't be accounted for in your gaseous measurements.

Line 332: "...which affects mainly the surface would influence the shrubs'....": I think it's important in this sentence to remind readers that surface changes affect shrubs and environmental changes deeper in the peat profile would affect sedges due to differences in the rooting depths of these two types of plants.

Line 350: Greater HR in drier periods may have more to do with the relationship between microbial activity and soil aeration/oxygen availability than the physiology of mosses and vascular plant/moss interactions.

Line 358: "...no difference between the two years in our study too though...": No difference in which variables?

Line 364: "...why the respiration values...": Which respiration values?

Line 376: "...they were no longer able to benefit from this priming effect..": But microbial activity should also be stimulated by rhizodeposition/priming from vascular plants...

Lines 385-386: "Especially with regards to unveiling the presence of the intermediate form of respiration we deemed plant-mediated HR..": Compared to the original version of this manuscript, all other mention of plant-mediated HR has been removed according to previous reviewer suggestions. Furthermore, I disagree completely that the authors have unveiled an intermediate contribution to ecosystem respiration. As stated above, the authors have no evidence to support the idea that mosses are essentially recycling $CO_2$ respired by microbes or by the surrounding vascular plants, and regardless, respiration stemming from mosses is not included in the category of heterotrophic respiration, which is entirely attributed to microbial activity.

Lines 390-393: The reference to soil nutrient dynamics seems odd here, if not totally unrelated to the rest of the information discussed at length in this manuscript. This is the first time soil nutrients are mentioned as a driver of soil respiration and seem out of place in a conclusion meant to summarize this manuscript's findings.

---

## Author Response (AR2)

**Responses to Reviewer 1:**

The revised manuscript has substantially improved the explanation of the methods and results, and the updated figures represent significant improvements in terms of data presentation and readability. I thought the new Figures S1 and S2 were particularly useful for showing the relationships among fluxes and key environmental drivers. The changes to the text and figures also clarify the rationale for separating the analysis of the two years. Overall, I think the revisions to the manuscript have addressed the concerns I raised in my previous review. My main remaining suggestion would be to include Figures S1 and S2 in the main text rather than supplementary material. These figures show very clearly the relationships with carbon fluxes, temperatures, and water table that are a focus of the manuscript, and they provide useful clarity when paired with the time series plots in Figures 3 and 4. I feel that the supplemental figures represent an important component of the manuscript's main message and therefore fit better in the main text.

*The authors thank the reviewer for their comments. We have changed Figures S1 and S2 to be Figures 6 and 7 as suggested and have included them in the text.*

In Figure S1 and S2, I would also suggest showing which points were from high or low WT periods in all the panels, not just the WT Depth panels. Showing those differences in the plots of fluxes vs. temperature would help to visualize some of the interactions between temperature and water table and make it easier to see how the different ranges of water table conditions coincided with temperature conditions.

*We have modified the figures which we hope assuages the concerns of the reviewer.*

Other specific comments:

Line 163: "anomalously warm year in many places across the globe" – I would say whether it was an anomalously warm year in this site specifically since other places on the globe don't directly impact these measured fluxes.

*Revised accordingly (line 174)*

Line 329-330: This conclusion suggests that showing the shrub only plots in scatter plots as in Figure S1 might be a useful addition.

*Revised accordingly to include a reference to Figures 6 and 7 (lines 340 - 341).*

**Responses to Reviewer 2:**

This manuscript provides a valuable dataset and analyses that describe carbon fluxes from a bog, portioned into the contributions to ecosystem respiration by autotrophic (AR) and heterotrophic (HR) sources. This dataset itself is unique, and the analyses that link respiration to the influence of water table depth, temperature, and plant community dynamics make valuable contributions to the field. The manuscript, namely the statistical analyses and figures, are much improved from the initial submission. However, I have remaining concerns regarding the interpretation of the results and the discussion section especially regarding the interactions between mosses and vascular plants that affect heterotrophic respiration rates (mainly the last paragraph of section 4.2, starting line 364).

*The authors thank the reviewer for their comments and will address each in turn.*

After reviewing the original manuscript, both reviewers raised concerns with the authors' discussion of symbiotic relationships between mosses and vascular plants in relation to HR rates, particularly the ability of mosses to metabolize and respire labile carbon from vascular plant root exudates. First, it seems incorrect to discuss respiration from mosses in the context of HR, which by definition includes only respiration from the soil microbial community. Second, the ability of mosses to absorb, metabolize, and respire dissolved carbon from the soil solution is not widely accepted, and the authors give a single source from 1999 to support these speculations. The authors simply do not have the data to make claims regarding the source of the carbon respired by plants in the experimental plots studied here. I suggest instead drawing speculations regarding the effect of mosses on heterotrophic respiration rates from known relationships in the literature that connect the presence of mosses to water table depth/soil moisture and temperature, variables that the authors show significantly shape HR. It seems plausible that mosses could insulate peat from evaporative losses of water, perhaps even establishing cooler, anaerobic conditions beneath mosses that limit microbial activity and HR when moss is present. When moss is removed, the albedo of the bog surface changes so that the bog surface would become warmer, and evaporation would occur more rapidly without the insulative presence of mosses to allow for more rapid aerobic microbial activity. These types of relationships, between mosses and microbial activity and the environmental factors that shape microbial activity, warrant further discussion in the manuscript segments concerning HR and might be more appropriate and scientifically sound than discussing moss respiration from dissolved carbon sources.

*We have provided a clearer explanation for the relationship between the mosses and the shrubs at Mer Bleue and provided citations of more recent research that show the possibility of the claims made in this study.*

Other minor concerns remain as well, as noted below:

Line 9: "respiration microbial bacteria in soil, fungi, etc.": "respiration by the soil microbial community" would probably be a more concise and accurate description of HR

*Revised accordingly (line 9)*

Line 11: "and alters allocations of carbon to labile pools with different turnover rates": The relationship between respiration and carbon substrate complexity was not something that was examined in this study and should be removed here and elsewhere in this manuscript to avoid confusing readers regarding the factors analyzed in this relationship. This argument distracts from the more evidencebased conclusions you make regarding the influence of abiotic factors and plant functional type on ecosystem respiration.

*Revised accordingly to remove statement (line 10)*

Line 20: Here and elsewhere in the manuscript, you discuss different plant water acquisition strategies as an important driver of respiration rates in this system, but what you actually discuss is differences among plant rooting structures; these differences may or may not be actually tied to water acquisition. I think you should refer to these differences as rooting structure differences instead of water acquisition differences because that more accurately describes the arguments you make in your discussion and the literature that you cite as you do not measure water acquisition in this study.

*Revised accordingly to describe the differences among rooting structures, but also stating that this indicates a difference in how the plants obtain water resources (line 20)*

Line 31: "the dynamics of heterotrophic respiration... is not straightforward" should be "are not straightforward"

*Revised accordingly (line 31)*

Line 32: "substrate variables": Can you be more specific? I think this covers a very broad spectrum of biogeochemical variables... Substrate quality or complexity? Substrate quantity?

*Revised accordingly to include examples (line 32)*

Line 36: "...and CO2 that is supplied as a substrate by vascular plants": This sentence is misleading, and the intended point here is unclear. CO2 gas is not provided by plants as a substrate for microbial metabolism.

*Revised accordingly to indicate "organic C supplied by plants" (line 37)*

Line 44: "Ecosystem Respiration dynamics...": the R in respiration should not be capitalized.

*Revised accordingly (line 44)*

Line 51: "For example, a positive feedback in climate change...": Several typos and/or confusing word choice make this sentence hard to follow and the intended point unclear.

*We rearranged the sentence structure for this paragraph (lines 51-54)*

 Line 53: "... turn over newly-photosynthesizing C..." should probably be "newlyphotosynthesized C"

*Revised accordingly (line 53)*

Lines 65-68: "This also indicates a problem in the conceptualization of ER: one cannot...": This sentence is very unclear and perhaps should be broken into several separate points. It's hard to understand what connection the authors are trying to make between CO2 released during HR and litter production as an intermediate contribution to HR... Carbon respired by the microbial community is still considered HR regardless of what the ultimate fate of those CO2 molecules are.

*We revised this paragraph to say there is a problem in our conceptualization of HR, not ER. While the typical definition of HR includes only respiration from the soil microbial community, as the reviewer*

*suggests above, we are claiming that HR is more complicated than that. Studies have suggested for a while now (citations provided in text) that HR is more related to vegetation dynamics and that root-soil interactions play a major role (lines 65-72).*

Line 83: Mean annual precipitation is listed as 943 mm, but this sentence states that annual snowfall is 223 cm. Are the units for snowfall accurate? I realize that there is some discrepancy for measured snowfall and realized water increments, but are those differences an order of magnitude apart?

*These are the reported values on Environment and Climate Change Canada's website for the Canadian climate normal from 1981-2010. Snow is measured in cm with a ruler, then precipitation is reported as the water equivalent from all sources in mm.*

Line 96: "...hence supporting a greater belowground biomass than sedges": This conclusion doesn't necessarily follow logically from the information you presented in this paragraph and at the very least, deserves its own reference. Why would the shallower root structure of shrubs necessarily equate to greater below ground biomass than the more deeply rooted sedges?

*Revised accordingly to explain it is the relative BG to AG biomass that is greater in shrubs than in sedges, not the magnitude of the biomass (line 96)*

Line 99: Live moss biomass still counts as aboveground biomass, so it seems illogical to consider stems buried by living moss to be considered belowground biomass.

*As mentioned in line 97 and 100-101, it is the stems of the shrubs that are buried with the help of the mosses, not the live moss biomass.*

Line 103: The conclusion made in the first sentence of this paragraph seems unsupported by the information provided. You state that sedges have a competitive advantage over shrubs yet start this sentence by stating that shrubs thrive in both dry and wet conditions. These statements seem to contradict one another.

*Revised accordingly to include that while shrubs are adaptable to a changing climate, sedges have a competitive advantage as they can handle more extreme fluctuations in soil moisture (lines 105-106).*

Line 110: Does the placement of the 9 collars represent 1 collar per plot? If so, how big are these plots within the shrub and sedge sections? As currently written, this seems to indicate pseudoreplication, where you measure respiration in 9 places with a single shrub and a single sedge plot.

*We restructured this section of the methods to be clearer on the manual plot set up (lines 110 - 126)*

Line 122: How big are each of the subplots with the vegetation manipulations?

*We revised accordingly to make this clearer (line 112).*

Line 127: Were the HR plots trenched to kill roots from plants that grow outside of the plots but close enough to produce roots that grow belowground into the plot area?

*Revised accordingly to clarify the root exclosure set up (lines 128-129)*

Line 139: "regression equations": linear models? Line 195: "Although it is important to acknowledge the hysteresis present...": I think you should describe why acknowledging hysteresis is important. It essentially means that WT depth is a better predictor of VWC in 2019 than in 2018.

*Revised accordingly to include linear regressions (line 142). We made a variety of scatterplots (not shown) to make sure there were no non-linear relationships between $CO_2$ concentrations over time.*

Section 3.2: Listing these results in the text is rather difficult to trudge through and derive any real sense of comparison between respiration rates in different years, with different plant manipulations, and between different plant types. Lit the results in a table instead, and use this section to describe comparisons between respiration rates among your treatments... i.e., Shrub respiration was x% greater in 2018 etc.

*We do not want to change this and refer to figures that clearly show the trends. However, we did compile the results in a table (table 2) and refer to the table in these paragraphs.*

Line 281: You mention "land-use" as a factor shaping respiration rates at many different points in this manuscript, but it is certainly not a factor that you explore in this experiment... I would advocate for removing reference to land-use from this manuscript not because it's not important, but because you don't investigate the influence of land use in your study, but you do investigate the influence of environmental and vegetative properties which should be your sole focus.

*We've removed all mention of land-use from the text as we agree with the reviewer that this was not studied.*

Line 320: Have you considered the fact that water table differences between your vegetation types alone might explain the vegetative differences in respiration rates? When the water table is higher, more deeply rooted plants will respire in totally saturated peat conditions, and a large portion of that respired CO2 will be dissolved in the soil solution instead of being released from the soil as gaseous CO2. This would mean that when the water table is higher, AR is automatically lower because at least a portion of deep root respiration won't be accounted for in your gaseous measurements.

*This is a good point made by the reviewer, but the fact that the interaction between temperature and moisture explains a large portion of the variability in respiration rates, suggests that water table depth is not the only factor.*

Line 332: "...which affects mainly the surface would influence the shrubs'....": I think it's important in this sentence to remind readers that surface changes affect shrubs and environmental changes deeper in the peat profile would affect sedges due to differences in the rooting depths of these two types of plants.

*Revised accordingly (lines 340 – 341)*

Line 350: Greater HR in drier periods may have more to do with the relationship between microbial activity and soil aeration/oxygen availability than the physiology of mosses and vascular plant/moss interactions.

*Revised accordingly (lines 344 – 345)*

Line 358: "…no difference between the two years in our study too though…": No difference in which variables?

*Revised accordingly (line 365)*

Line 364: "…why the respiration values…": Which respiration values?

*Revised accordingly (lines 371 - 372)*

Line 376: "…they were no longer able to benefit from this priming effect.": But microbial activity should also be stimulated by rhizodeposition/priming from vascular plants…

*This sentence does not imply that no microbial activity is occurring; we suggest that the mosses are not benefiting from the change in vegetation cover.*

Lines 385-386: "Especially with regards to unveiling the presence of the intermediate form of respiration we deemed plant-mediated HR..": Compared to the original version of this manuscript, all other mention of plant-mediated HR has been removed according to previous reviewer suggestions. Furthermore, I disagree completely that the authors have unveiled an intermediate contribution to ecosystem respiration. As stated above, the authors have no evidence to support the idea that mosses are essentially recycling $CO_2$ respired by microbes or by the surrounding vascular plants, and regardless, respiration stemming from mosses is not included in the category of heterotrophic respiration, which is entirely attributed to microbial activity.

*Revised accordingly to refer to plant-associated HR rather than plant-mediated HR (line 396). And as stated above, we provided more evidence of the vegetation influence on HR in the text. We suggest this is plausible and that our conceptualization of HR needs to be changed (e.g., lines 386 – 389).*

Lines 390-393: The reference to soil nutrient dynamics seems odd here, if not totally unrelated to the rest of the information discussed at length in this manuscript. This is the first time soil nutrients are mentioned as a driver of soil respiration and seem out of place in a conclusion meant to summarize this manuscript's findings.

*We were suggesting something future studies can explore since it has been shown in many studies that moisture and nutrient dynamics are linked.*

---

## Author Response (AR3)

**Author's Response – Publication Submission**

**Reply to Editor**

Dear Authors,

I am pleased to recommend publication subject to two minor technical corrections that the Reviewer has noted, please could you address this in your revised manuscript (see below).

Best wishes,

Martin De Kauwe

*The Author's thank the editor for his recommendation to publish the manuscript. We have completed the minor corrections the reviewer suggested. We also corrected the Helbig et al. 2013 reference in lines 505 - 508 where the superscripts for the carbon isotopes in the title were not correct as well as removed the space at line 512.*

**Reply to Reviewer**

"The manuscript reads well and my concerns on previous versions have been addressed. I did notice that while the legend on Figures 6 and 7 now include separate symbols for high water table and low water table in 2018, the low water table symbol (blue diamond) does not appear in panels g, h, and i of Figure 6 and does not appear at all in Figure 7. This seems like a mistake in the plots because low water table is defined as below 35 cm and water tables below that level do appear in Figure 7 but are marked with blue circles instead of diamonds.

*The authors have made the minor corrections to the figures and thank the reviewer for their comment.*

Also, Figure 3 has a "Plot Area" box on it that appears to be an artifact from the plotting program."

*The "plot area" box was removed from the figure accordingly.*